# Canine uropathogenic and avian pathogenic *Escherichia coli* harboring conjugative plasmids exhibit augmented growth and exopolysaccharide production in response to *Enterococcus faecalis*

**Grayson K. Walker** [ID]\***, M. Mitsu Suyemoto, Megan E. Jacob, Siddhartha Thakur** [ID]**, Luke B. Borst**

Department of Population Health and Pathobiology, College of Veterinary Medicine, North Carolina State University, Raleigh, NC, United States of America

\* gkwalke2@ncsu.edu

## Abstract

Uropathogenic *Escherichia coli* (UPEC) and avian pathogenic *Escherichia coli* (APEC) are extraintestinal pathogenic *Escherichia coli* (ExPEC) that infect dogs and poultry. These agents occur both as single-species infections and, commonly, in co-infection with *Enterococcus faecalis* (EF); however, it is unclear how EF co-infections modulate ExPEC virulence. Genetic drivers of interspecies interactions affecting virulence were identified using macrocolony co-culture, chicken embryo co-infection experiments, and whole-genome sequence analysis of ExPEC and EF clinical isolates. Ten of 11 UPEC strains originally co-isolated with EF exhibited a growth advantage when co-cultured with EF on iron-limited, semi-solid media in contrast to growing alone ($P < 0.01$). Phylogenetic analyses of these UPEC and 18 previously screened APEC indicated the growth-response phenotype was conserved in ExPEC despite strain diversity. When genomes of EF-responsive ExPEC were compared to non-responsive ExPEC genomes, EF-induced growth was associated with siderophore, exopolysaccharide (EPS), and plasmid conjugative transfer genes. Two matched pairs of EF-responsive and non-responsive ExPEC were selected for further characterization by macrocolony proximity and chicken embryo lethality assays. EF-responsive ExPEC produced 5 to 16 times more EPS in proximity to EF and were more lethal to embryos alone and during co-infection with EF compared to non-responsive ExPEC ($P < 0.05$). A responsive APEC strain cured of its conjugative plasmid lost the enhanced growth and EPS production response to EF. These data demonstrate that ExPEC growth augmentation by EF occurs in UPEC and APEC strains and is linked to conjugative virulence plasmids and EPS production, which are widely conserved ExPEC virulence determinants.

**Data Availability Statement:** All relevant data are within the paper and Supporting Information files. Whole genome sequence data are available from the GenBank database (BioProjects PRJNA293225 and PRJNA837978). Accession numbers for each strain are listed in Table 1 and Figs 2 and 3.

**Funding:** The whole-genome sequencing was completed by the U.S. Food and Drug Administration (FDA) GenomeTrakr program-funded grant 1U18FD00678801. Additional funding was provided by the United States Department of Agriculture (USDA) Animal Plant Health Inspection Service National Bio and Agro-Defense Facility Scientist Training Program. The funders had no role in study design, data collection and analysis, decision to publish, or preparation of the manuscript.

**Competing interests:** The authors have declared that no competing interests exist.

## Introduction

Extraintestinal pathogenic *Escherichia coli* (ExPEC) are ubiquitous pathogens and a leading cause of morbidity and mortality resulting from extraintestinal infections among humans and animals [1]. In veterinary medicine, extraintestinal infections such as canine urinary tract infections (UTI) and avian colisepticemia are caused by the ExPEC pathovars uropathogenic *E. coli* (UPEC) [2] and avian pathogenic *E. coli* (APEC) [3], respectively. UPEC are the leading cause of bacterial cystitis and recurrent UTI in domestic dogs and humans and can be zoonotically or anthroponotically transmitted [2, 4]. APEC spread rapidly among intensively reared chickens and turkeys causing severe systemic infections and high mortality that result in multi-million-dollar annual losses in the poultry sector [3]. ExPEC pathovars UPEC and APEC are frequently implicated in polymicrobial extraintestinal infections of their respective hosts, which further complicate diagnosis and treatment [4–6]. Epidemiological and experimental studies have revealed that *Enterococcus faecalis* (EF) are frequently implicated in polymicrobial ExPEC infections and can augment ExPEC virulence [7–9]. However, genetic factors driving EF and ExPEC interactions that induce ExPEC virulence and disease progression are unclear.

EF form robust biofilms and have intrinsic antimicrobial resistance and immune evasion features making them well-adapted as hospital-acquired, extraintestinal pathogens [7, 10]. Genetically related, multidrug-resistant EF lineages are often transmitted in healthcare and household settings [11–13]. In contrast, ExPEC represent a heterogeneous pathogen population that relies on diverse virulence mechanisms to escape the intestinal niche and cause systemic extraintestinal disease [14]. Multiple and often synergistic ExPEC virulence mechanisms for survival in the host extraintestinal niche have been identified and include siderophore-mediated iron acquisition and production of exopolysaccharides (EPS) such as lipopolysaccharide (LPS) and colanic acid (CA) [15, 16]. Iron is a limiting factor for ExPEC metabolism and is restricted by host nutritional immunity in the extraintestinal environment [17]. EPS are produced in response to low iron and protect ExPEC from antimicrobial peptides and complement-mediated killing [18, 19]. These features make EPS an important virulence feature that ExPEC utilize for systemic colonization of animal hosts [19].

Interestingly, co-culture with EF appears to alleviate growth impairments in ExPEC due to iron-limitation [5, 8]. A mechanism by which EF secretes L-ornithine to augment ExPEC siderophore activity has been described [8] but studies investigating the prevalence of EF interactions with ExPEC isolated from different hosts and disease processes are lacking. In a previous study, we screened 31 APEC strains for growth enhancement by EF in mixed culture on solid, iron-restricted medium to simulate an *in vivo* extraintestinal microenvironment [5]. In these conditions, 30 of 31 APEC strains exhibited robust growth augmentation in response to EF. Further, co-infection of APEC and EF resulted in increased virulence in chicken embryos compared to single-species infections [5]. As these findings were highly conserved in APEC isolates, the objectives of this work were: 1) to assess UPEC strains from canine UTI for growth enhancement in the presence of EF in iron-deplete media and 2) identify potential genetic drivers of EF induced growth and virulence mechanisms in ExPEC using whole-genome sequencing and comparative genomic analysis combined with *in vitro* and *in vivo* models of co-infection.

## Results

### Canine uropathogenic *E. coli* macrocolony growth augmentation

A collection of 11 uropathogenic *E. coli* (UPEC) isolated from dogs diagnosed with urinary tract co-infection of UPEC and *Enterococcus* spp. were screened for growth augmentation by

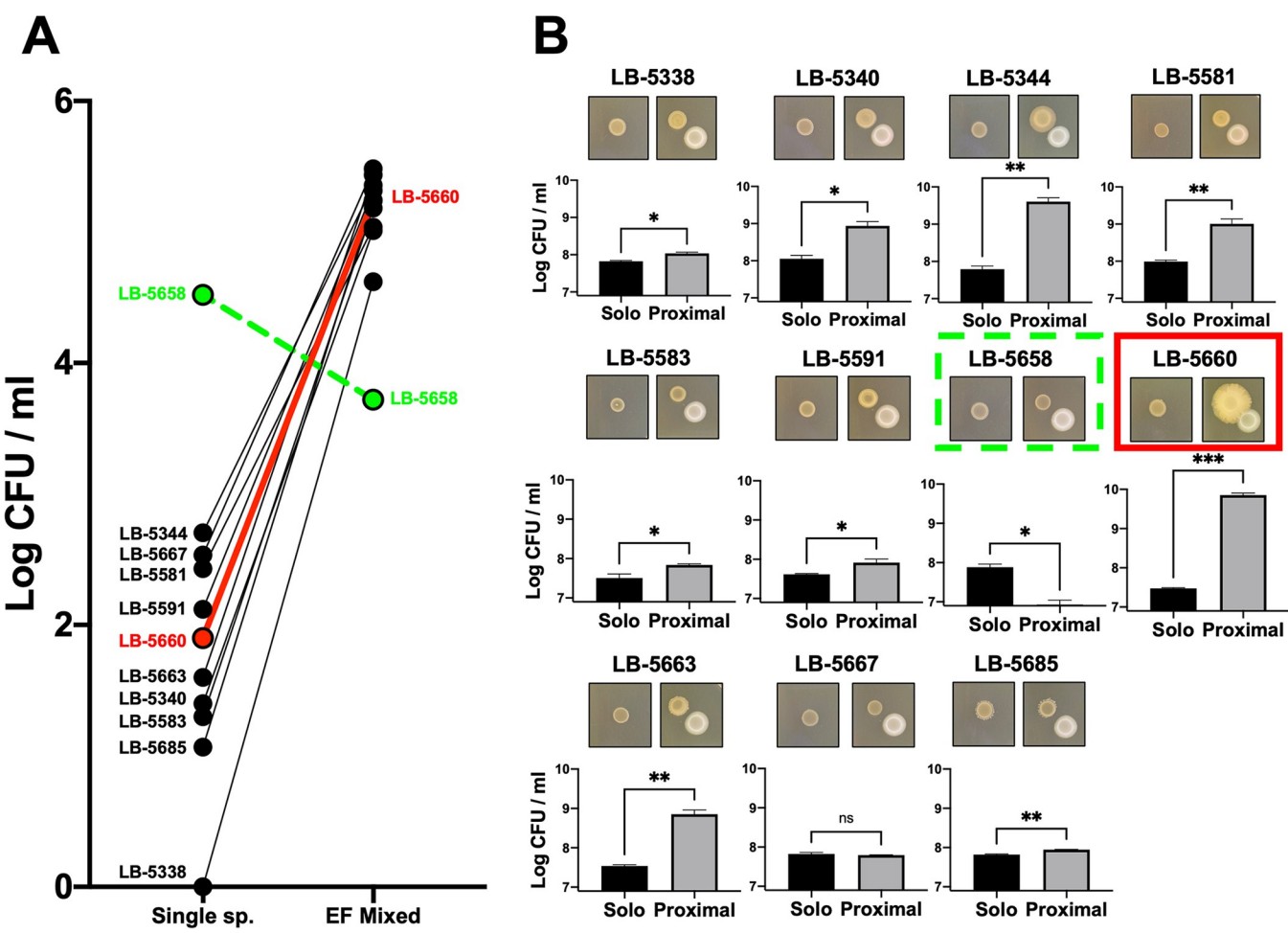

**Fig 1. Growth induction of uropathogenic *Escherichia coli* (UPEC) isolates from dogs by *Enterococcus faecalis* OG1RF (EF).** (A) Growth of 11 UPEC strains in single species macrocolonies or mixed with EF after 24 hr on trypticase soy agar with 10 mM glucose (TSA + G) and 1.0 mM iron chelator 2,2'-dipyridyl (22D). Co-isolated strains LB-5660 (solid red) and LB-5658 (dashed green) were designated as EF-responsive and non-responsive, respectively, and further characterized. (B) Macrocolony growth in proximity to EF of 11 UPEC, including LB-5658 (dashed green box) and LB-5660 (solid red box), that were co-isolated with *Enterococcus* spp. from canine urine samples. UPEC macrocolonies were grown for 5 days on TSA + G with 0.15 mM 22D either in isolated quadrants or 1 cm from an EF macrocolony, excised, and their growth quantitated by serial dilution and plating to obtain colony forming units (CFU) / ml values. *$P \leq 0.05$, ** $P \leq 0.01$, *** $P \leq 0.001$, ns = $P > 0.05$.

*Enterococcus faecalis* (EF) (Fig 1). Growth of UPEC macrocolonies alone was compared to growth in mixed culture with EF at two concentrations of iron chelator on semi-solid media. First, 11 UPEC were screened for an enhanced growth response to EF in mixed culture on medium with a high 1.0 mM concentration of the iron chelator 2,2'-dipyridyl (22D). Under stringent iron limitation, all but one UPEC strain LB-5658 had significantly enhanced growth in mixed culture ranging from 2.60 to 4.62 Log colony forming units (CFU) when compared to single species macrocolonies ($P < 0.01$, Fig 1A).

To assess the growth response at a less stringent level of iron restriction, UPEC strains were further investigated using a macrocolony proximity assay with a reduced level of iron chelator (0.15 mM 22D). Growth enhancement of 11 UPEC macrocolonies grown 1 cm from EF for 5 d corresponded to the growth responses observed in mixed culture with EF and stringent iron chelation and allowed for macrocolony phenotype visualization (Fig 1B). LB-5658, which had inhibited growth in mixed culture with EF, was also inhibited in proximity to EF in the

macrocolony proximity assay (Fig 1A and 1B). In contrast, strain LB-5660 demonstrated a 3.34 Log growth increase grown under high iron restriction in mixed culture with EF and a 2.4 Log growth response in less stringent iron restriction in proximity to EF (*P* < 0.001, Fig 1A and 1B). These methods were previously used to establish the EF-induced growth response phenotypes of avian pathogenic *E. coli* (APEC) strains [5]. In this study, NC_LB-5134 demonstrated a 5.01 Log growth increase grown under high iron restriction in mixed culture with EF and robust macrocolony size increase when grown 1 cm from EF under less stringent iron restrictions while strain LB-4801 lacked a significant response in either condition.

**Molecular typing and phylogenetic analysis of co-isolated *E. coli* and *Enterococcus faecalis*.** Phylogenetic trees were constructed in tandem for a collection of APEC and EF co-isolated from poultry with colisepticemia (n = 18 per species). Whole genome single nucleotide polymorphism (SNP) analysis and phylotyping revealed heterogeneity among co-isolated APEC and EF strains (Fig 2). APEC strains clustered into 16 unique SNP groups and all but phylogroup C of the 7 Clermont phylogroups were represented. Phylogroups A (3/19), B1 (5/19), the extraintestinal pathogenic *E. coli* (ExPEC)-associated B2 (3/19), and D (3/19) were most abundant among APEC. EF isolates clustered into 13 unique SNP groups and 7 unique multilocus sequence types (MLST). Four of the EF strain lineages were not typable by MLST. A phylogenetic tree was then constructed for the collection of UPEC strains screened above. Similar to the APEC co-isolates, UPEC co-isolates represented 9 of 11 unique SNP groups with the B2 phylogroup highly represented (5/11) but phylogroups D, E, and F were absent (Fig 3). These results indicated that APEC and UPEC co-isolated with EF from extraintestinal infections did not belong to distinct lineages.

## Comparative genomic analysis of responding *E. coli* strains

The genome annotations of representative APEC and UPEC strains that exhibited growth and macrocolony size augmentation by EF were compared to representative strains from each group that lacked this response. EF-responsive ExPEC strains with robust macrocolony growth responses included NC_LB-5134 and LB-5660 (Fig 1B). Genome annotations and plasmid replicon types from these strains were compared to those of LB-5658 and LB-4801 which did not show increased growth in mixed culture with EF (Table 1, Fig 5). Functional genome annotations revealed that EF-responsive NC_LB-5134 and LB-5660 had 58 and 75 predicted proteins, respectively, that were absent in the non-responsive LB-5658 and LB-4801 reference pair (S1 Table). The IncFIB plasmid replicon (NCBI accession number AP001918) and genes encoding conjugative transfer pilin were absent in LB-5658 but present in the three other strains (Table 1). Additionally, the aerobactin and yersiniabactin siderophores and capsular colonic acid (CA) biosynthesis genes were unique features of both LB-5660 and NC_LB-5134 but absent in LB-5658 and LB-4801 (Table 1, Fig 4). The discovery of these conserved features in EF-responsive *E. coli* strains prompted measurement of these phenotypes using the *in vitro* macrocolony co-culture assay.

## *E. coli* growth and exopolysaccharide production in response to *E. faecalis*

ExPEC strains LB-5660 and NC_LB-5134 each exhibited a proximity-dependent increase in macrocolony size and mucoid appearance in response to EF (Fig 5A). This increase in macrocolony diameter corresponded to over 2.5 Log more growth when compared to ExPEC grown in single species culture (*P* < 0.01, Fig 5B). LB-4801, which did not show increased growth in mixed culture with EF on the high iron chelator medium, had a 1 Log growth increase in response to EF in the proximity assay on the lower 22D medium (*P* < 0.01, Fig 5B); however, the mucoid phenotype was not observed (Fig 5A). Since LB-5660 and NC_LB-5134 had genes

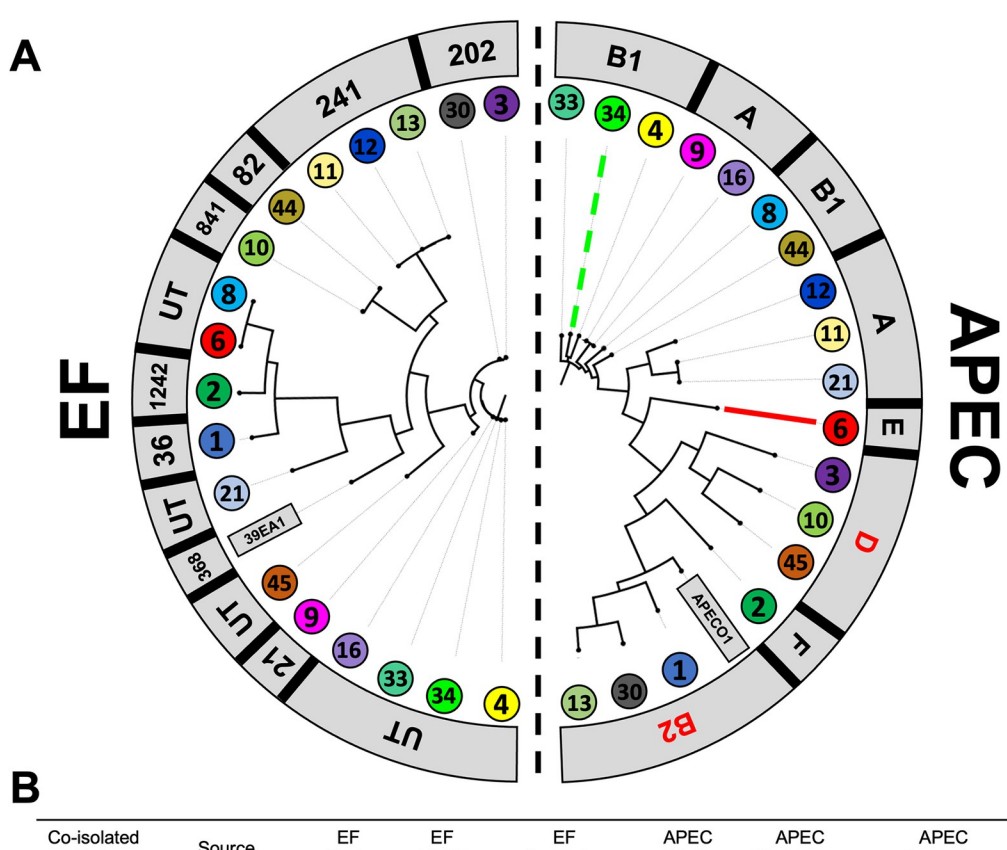

**Fig 2. Molecular typing and phylogenetic analysis of *Enterococcus faecalis* (EF) and avian pathogenic *Escherichia coli* (APEC).** (A) Whole-genome single nucleotide polymorphism (SNP) analysis of APEC and EF co-isolated from poultry diagnosed with colisepticemia aligned to the APECO1 (NCBI accession: NC_008563.1) and *E. faecalis* 39EA1 (NCBI accession NZ_KZ846041.1) reference genomes, respectively. Co-isolated pairs are indicated by the same-colored number and Clermont phylogroups of APEC are indicated in grey boxes with groups B and D2 associated with pathogenic *E. coli* depicted in red. EF multilocus sequence types (MLST) are indicated in grey boxes. EF strains that had unknown MLST patterns are designated as untypable (UT). APEC strains selected for further phenotypic and genomic characterization included NC_LB-5134 (from co-isolated pair 6, solid red line) and LB-4801 (from co-isolated pair 34, dashed green line) which exhibited growth augmentation by EF or no growth response. (B) Inset table depicting EF and APEC strain metadata and NCBI accession numbers for draft genomes used for phylogenetic analysis. APEC strains NC_LB-5134 and LB-4801 are depicted in solid red and dashed green boxes, respectively.

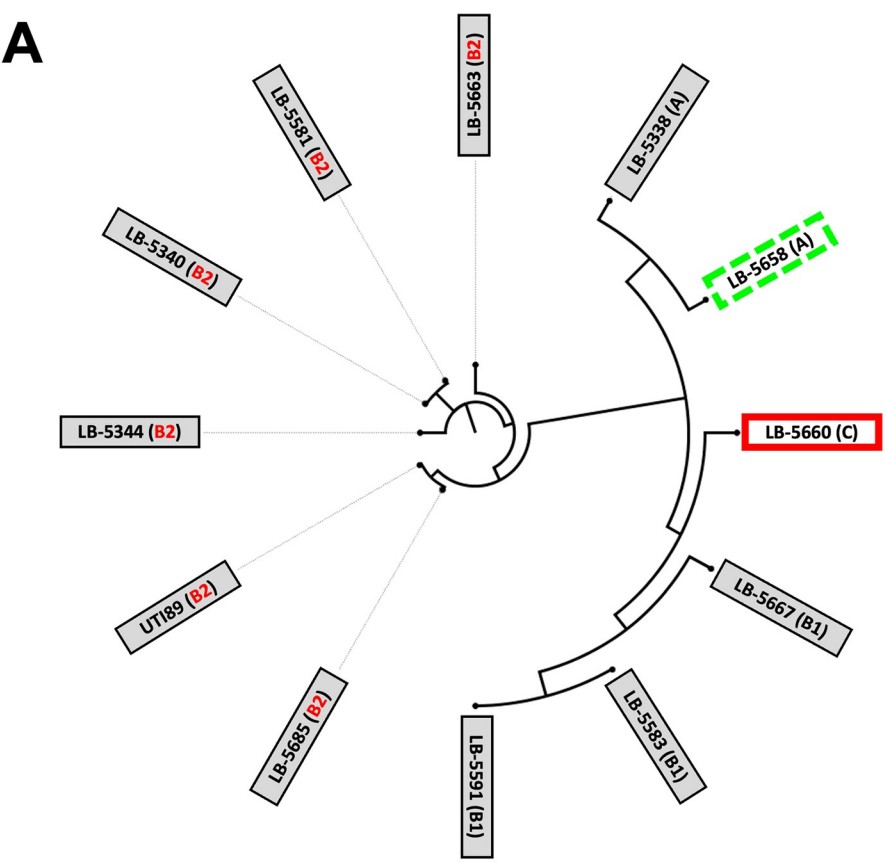

**Fig 3. Molecular typing and phylogenetic analysis of canine uropathogenic *Escherichia coli* (UPEC).** (A) Whole-genome single nucleotide polymorphism (SNP) analysis of UPEC co-isolated with *Enterococcus* spp. from dogs diagnosed with urinary tract co-infections aligned to the UTI89 reference genome (NCBI accession NC_007946.1) with Clermont phylogroups in parenthesis. UPEC strains selected for further phenotypic and genomic characterization included LB-5660 (solid red box) and LB-5658 (dashed green box) which exhibited growth augmentation by EF or no growth response as determined by this study (B) Inset table depicting UPEC strain metadata and NCBI accession numbers for draft genomes used for phylogenetic analysis. UPEC strains LB-5660 and LB-5658 are depicted in solid red and dashed green boxes, respectively.

**Table 1. Comparison of extraintestinal pathogenic *Escherichia coli* (ExPEC) strains NC_LB-5134 and LB-5660 with a robust growth response to *Enterococcus faecalis* OG1RF (EF) as compared to unresponsive ExPEC strains LB-4801 and LB-5658.**

| | Strain | | | |
|---|---|---|---|---|
| | **NC_LB-5134** | **LB-4801** | **LB-5660** | **LB-5658** |
| GenBank Accession | ABFMQO000000000 | AAUEYM00000000 | ABAXNK000000000 | ABAXNM000000000 |
| Morphology[1] | Smooth | Rough | Smooth | Rough |
| Response to EF[2] | High | Low | High | Low |
| O-antigen[3] | O131 | Untypable | Untypable | Untypable |
| MLST[4] | 57 | 2280 | 410 | 10 |
| Plasmid Replicon[5] | IncFIB | IncFIB (H89-PhagePlasmid), IncI1-I (Alpha) | IncFIB, IncFII (pHN7A8), Inc1-I (Alpha), IncP1 | IncHI1 |
| Conjugative Transfer *tra* Genes[6] | *traA-X* | *traA-X* | *traA-X* | Not detected |
| Aerobactin[6] | *iucD, iucC, iucB, iucA* | Not detected | *iucD, iucC, iucB, iucA* | Not detected |
| Yersiniabactin[6] | *irp1, irp3, ybtT* | Not detected | *irp1, irp3, ybtT* | Not detected |
| Colanic Acid Biosynthesis[6] | *wcaL, wcaK, wzxC, cpsB, wcaI, gmm, wcaE, wcaD* | Not detected | *wcaL, wcaK, wzxC, wcaJ, cpsB, wcaI, wcaD, wcaC* | Not detected |

[1]Colony morphologies were described after 24 hr growth at 37˚C on trypticase soy agar (TSA) with 5% sheep blood. Smooth colonies were round and slightly raised with smooth margins. Rough colonies were flat with rough, irregular margins.

[2]Response to EF was determined for each strain based on growth in mixed culture with EF or lack thereof after 24 hr on TSA with 10 mM glucose (TSA + G) and 1.0 mM iron chelator 2,2'-dipyridyl (22D).

[3]O-antigen typing was completed *in silico* by uploading genome assemblies to SerotypeFinder version 2.0 from the Center for Genomic Epidemiology.

[4]Multilocus sequence typing (MLST) was completed *in silico* by uploading genome assemblies to MLST version 2.0 from the Center for Genomic Epidemiology.

[5]Plasmid replicons with greater than or equal to 98% identity were detected with PlasmidFinder version 2.1 from the Center for Genomic Epidemiology.

[6]Genes were identified by comparing or manually searching functional genome annotations of NC_LB-5134, LB-4801, LB-5660, and LB-5658 in the RAST (Rapid Annotation using Subsystems Technology) web package. Sequence-based comparisons were made between the two pairs using the APECO1 reference genome and predicted proteins specific to ExPEC with a high response to EF were identified by filtering annotated genes with 98% similarity to reference sequences in the high response groups but were undetectable in the low response groups. PCR for the *traA* gene was confirmed by PCR.

required for CA biosynthesis and expressed mucoid phenotypes proximal to EF when compared to the non-responsive strains, exopolysaccharides (EPS) were quantitated from *E. coli* macrocolonies and normalized to cell density. Both EF-responsive LB-5660 and NC_LB-5134 produced more EPS when grown proximally to EF ($P < 0.05$) while LB-5658 and LB-4801 produced less EPS regardless of growth conditions (Fig 5C). These findings linked the ExPEC growth augmentation and macrocolony morphology changes to EPS production in response to EF.

## Embryo lethality assay

The contributions of these phenotypic changes to virulence during co-infection with EF strain OG1RF were investigated using a chicken embryo lethality assay. *E. coli* strains LB-5658 and LB-4801 had relatively low virulence to embryos during single-species infection but co-infection with EF resulted in significantly increased mortality ($P < 0.05$, Fig 5A and 5B). Single-species infections with LB-5660 and NC_LB-5134 were more virulent than LB-5658 and LB-4801 co-infections ($P < 0.05$, Fig 6A and 6B). Mortality did not increase with co-infection of EF and LB-5660 or NC_LB-5134 as compared to single species infection of these two responding *E. coli* strains.

## Plasmid genes and curing

ExPEC strains LB-5660, NC_LB-5134, and LB-4801 harbored the conjugative IncFIB plasmid (Table 1). Manual examination of the genome annotations and PCR gene amplification

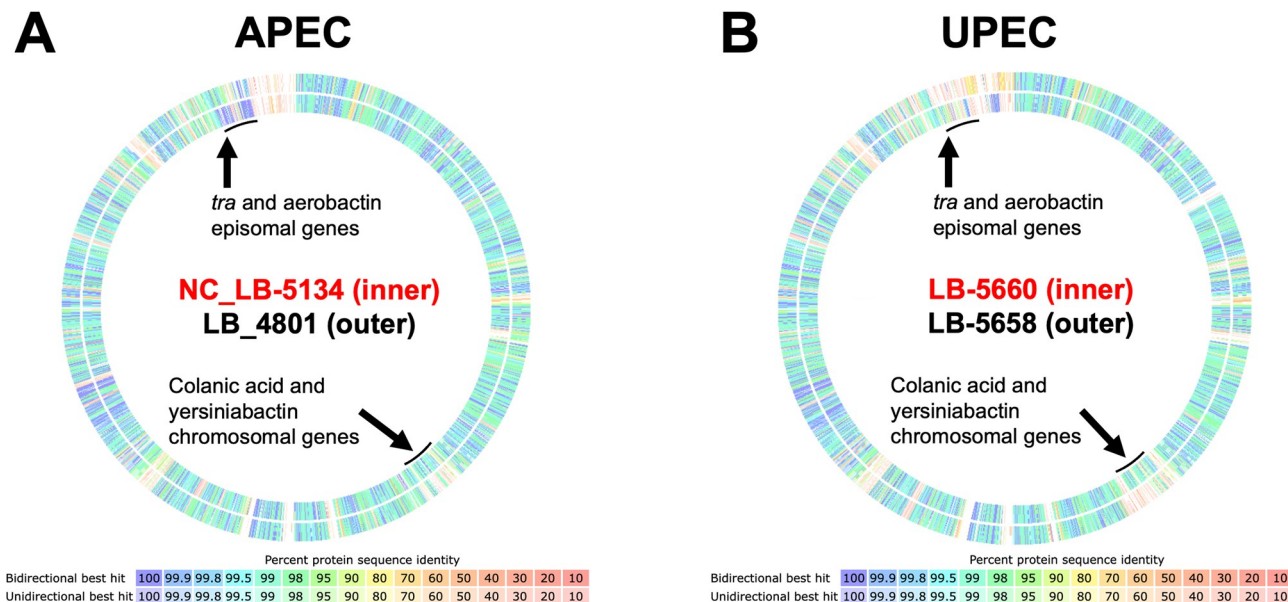

**Fig 4. Comparative genomic analysis of *Escherichia coli* strains with opposing growth augmentation phenotype responses to *Enterococcus faecalis* (EF) using the Rapid Annotation using Subsystems Technology (RAST) compare genomes feature.** Regions with predicted proteins for colanic acid and yersiniabactin biosynthesis, conjugative transfer (Tra) pilin and aerobactin are depicted for each strain combination. (A) Genome comparison of avian pathogenic *Escherichia coli* (APEC) strains: EF responsive NC_LB-5134 (red text, inner ring) to EF non-responsive LB-4801 (outer ring). (B) Genome comparison of canine uropathogenic *E. coli* (UPEC) strains: EF responsive LB-5660 (red text, inner ring) and EF non-responsive LB-5658 (outer ring).

indicated that all strains except LB-5658 had predicted conjugative transfer Tra proteins including TraA for pilin production (Table 1). The aerobactin siderophore identified in the comparative genomic analysis also localized to plasmid regions of LB-5660 and NC_LB-5134 but not LB-5658 or LB-4801 (Fig 4). Curing NC_LB-5134 of its IncF plasmid abolished EF proximity-dependent increases in macrocolony size, growth, and EPS production when compared to the wild type progenitor strain (Fig 7). When the embryo lethality assay was repeated with NC_LB-5134 and the plasmid-cured mutant strain, the virulence did not change in either the single strain infection or during co-infection with EF strain OG1RF (Fig 7D).

## Discussion

Extraintestinal pathogenic *E. coli* (ExPEC) are ubiquitous pathogens that cause severe systemic infections in humans and veterinary species. In veterinary medicine, dogs with UTI and poultry with colisepticemia are among the most commonly affected groups [2, 3]. A growing body of epidemiological and experimental evidence indicates that ExPEC infections may be potentiated by *Enterococcus faecalis* (EF) as ExPEC and enterococcal species are commonly co-isolated in polymicrobial extraintestinal infections [5–8]. However, strain-dependent interactions and genetic drivers of these interactions are unclear [5]. To better understand these interactions, we investigated relatedness and growth induction among two collections of avian pathogenic *E. coli* and uropathogenic *E. coli* co-isolated with EF. We discovered that *E. coli* plasmids and EPS production were implicated in the interaction using molecular typing and comparative genomics. Findings were tested using *in vitro* co-culture models and an *in vivo* embryo co-infection model to link these genotypes to phenotypic features of ExPEC including growth, exopolysaccharide (EPS) production, and virulence to chicken embryos.

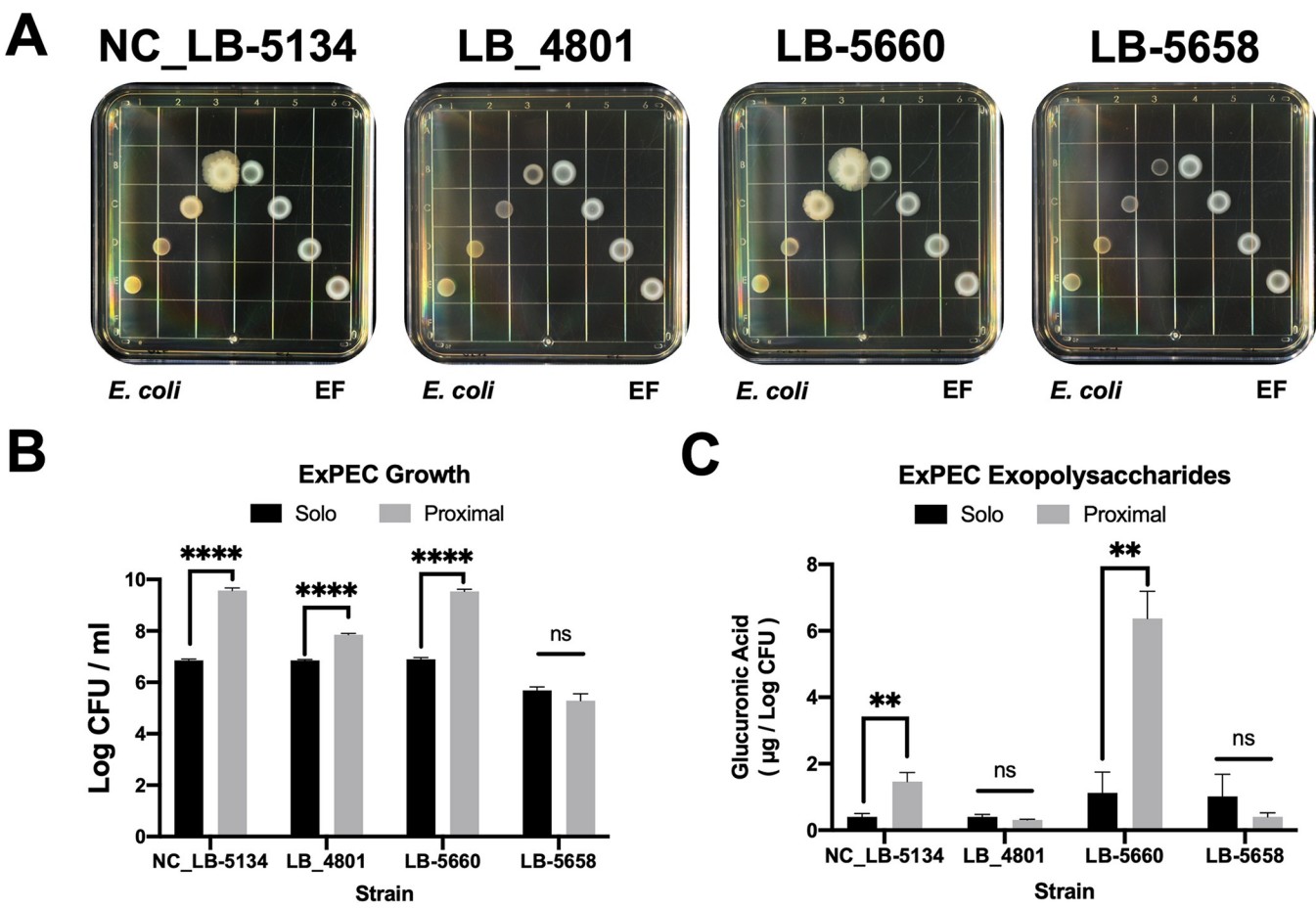

**Fig 5. Macrocolony phenotypes, growth, and exopolysaccharide (EPS) production by extraintestinal pathogenic *Escherichia coli* (ExPEC) strains NC_LB-5134, LB-4801, LB-5660, and LB-5658.** (A) Macrocolony proximity assay of ExPEC strains (left) grown in increasing proximity to *Enterococcus faecalis* OG1RF (EF, right) after 5 d on trypticase soy agar with 10 mM glucose (TSA + G) and 0.15 mM iron chelator 2,2'-dipyridyl (22D). (B) Macrocolony growth of each ExPEC strain in isolated single-species culture (solo) or 1 cm from EF (proximal) after 5 d on TSA + G and 0.15 mM 22D. (C) Total EPS production of each ExPEC strain, normalized to cell density. EPS was estimated by quantitating glucuronic acid using the method of Ma et al. [19]. ** $P \leq 0.01$, **** $P \leq 0.0001$, ns = $P > 0.05$.

After observing that EF-induced ExPEC survival under stringent iron limitation was a conserved feature of APEC [5], we were curious if this translated to UPEC isolated from canine UTI. Consistent with previous observations with APEC, nearly all UPEC strains demonstrated augmented growth in mixed culture with EF. Also similar to previous findings, UPEC isolates demonstrated genotypic diversity and clonal EF lineages were rare suggesting this growth-synergy during co-infection is a common and widespread phenomenon and not limited to distinct ExPEC or EF strains. Although ExPEC-associated phylogroups B2 and D were overrepresented among UPEC and APEC strains, other environmental and commensal *E. coli*-associated phylogroups were observed as well (Figs 2 and 3). As this interaction was widely conserved, we were interested in the outlier strains from each collection that did not express this EF-responsive phenotype, specifically how their genotypes differed from responsive strains and how that might translate to ExPEC virulence induction by EF during co-infections.

The functional genome annotations of ExPEC with robust growth responses to EF were compared to strains without growth induction by EF to identify active subsystems implicated in these phenotypes. Among the predicted proteins present in EF-responsive NC_LB-5134

and LB-5660 but absent in non-responsive LB-4801 and LB-5658 were siderophores yersinia-bactin and aerobactin and colanic acid (CA) biosynthesis genes. ExPEC utilize up to 4 sidero-phores: enterobactin, yersiniabactin, aerobactin, and salmochelin to sequester iron from extraintestinal tissues where access to this essential nutrient is severely limited [17]. These side-rophores are found in different combinations in ExPEC and function independently in response to environmental factors such as temperature and pH [20]. A mechanism by which pH changes and L-ornithine flux by EF modulate UPEC biosynthesis of enterobactin and yer-siniabactin has been described [8], yet EF effects on ExPEC expression of specific siderophores and their role in iron deplete conditions and during infection are unclear. Our findings suggest that aerobactin and/or yersiniabactin, in addition to the native *E. coli* siderophore, enterobac-tin, are associated with augmented ExPEC responses to EF when iron is limited. These acces-sory siderophores are ExPEC virulence determinants [21–24] and may be activated to different degrees in response to environmental conditions created by EF and the host response during co-infection [8, 20].

Also of interest was the association of capsular CA production with ExPEC growth induc-tion by EF (Table 1). CA production is common among ExPEC, and it is clear this EPS is not only required for *E. coli* biofilm architecture [25] but is also required for ExPEC extraintestinal colonization [26]. Because EPS protects ExPEC from antimicrobial immune defenses in extra-intestinal tissues [19] and CA-producing strains exhibited mucoid phenotypes in proximity to EF on a solid surface compared to CA deficient strains (Figs 1B and 5A), we chose to investi-gate EPS production of these different ExPEC strains in response to EF. Consistent with CA genotype disparities among the 4 ExPEC strains studied, only those with intact CA biosynthe-sis operons overproduced EPS in proximity to EF, independent of growth (Fig 5C). Interest-ingly, CA-deficient ExPEC also had impairments in LPS biosynthesis as they had untypable O-antigens and rough colony morphologies on blood agar (Table 1) [27]. Potential CA modula-tion by EF is an interesting finding as CA was required for ExPEC septicemia in a chicken infection model [16] and production of this EPS is regulated in response to low iron availabil-ity [19]. It was unclear if the absence of CA genes in non-responsive strains was an artifact related to other EPS synthesis impairments, such as LPS [27, 28]. Nevertheless, the finding that EF may potentiate ExPEC virulence mechanisms by increasing EPS warrants further investigation.

EPS production and iron acquisition are interlinked at the transcriptional regulatory level. Low iron relieves repression of EPS biosynthesis genes by the ferric uptake regulator, *fur* [19]. Thus, iron acquisition and EPS production work synergistically to both promote growth and virulence in the extraintestinal niche [19, 29]. Because EF induced ExPEC growth and EPS production in strains containing yersiniabactin, aerobactin, and intact CA biosynthesis oper-ons, we hypothesized these strains would be more lethal to chicken embryos during co-infec-tion compared to the ExPEC strains that lacked these features. ExPEC strains LB-5660 and NC_LB-5134 were more virulent alone and during co-infection with EF (Fig 6). However, the ExPEC strains lacking these features significantly benefitted from co-infection with EF despite low virulence alone and deficits in EPS production (Fig 6). This suggests an additional mecha-nism for virulence induction by EF in ExPEC that may not be linked to growth or EPS produc-tion. Similarly, the growth of nonpathogenic, LPS-deficient *E. coli* strains was augmented by EF in a murine catheter-associated UTI model of infection [7]. These findings support the hypothesis that EF increases the pathogenicity of otherwise commensal *E. coli* through unknown mechanisms.

The aerobactin siderophore is an episomal virulence feature in ExPEC [30] and conjugative transfer proteins encoded by F plasmids were features of ExPEC strains with a robust response to EF *in vitro*. Therefore, we next interrogated the role of F plasmids in this interaction. F

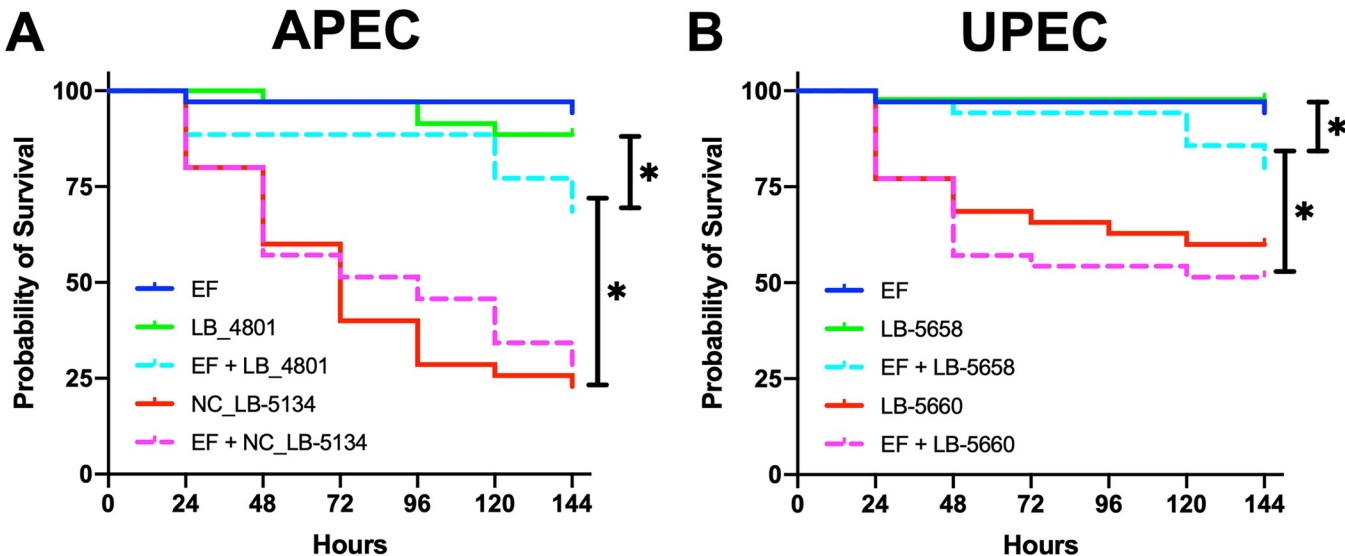

**Fig 6. Survival of chicken embryos after challenge with extraintestinal pathogenic *Escherichia coli* (ExPEC) strains NC_LB-5134, LB-4801, LB-5660, and LB-5658 alone or mixed with *Enterococcus faecalis* OG1RF (EF).** In each experiment, 35 embryos were challenged with approximately 2 x 10$^2$ CFU/egg of ExPEC strains alone or mixed with 4 × 10$^3$ CFU of EF via the intra-allantoic route at 12 d of incubation and candled every 24 hr to monitor viability. (A) Single species and co-infections of embryos with avian pathogenic *E. coli* (APEC) strains NC_LB-5134 and LB-4801. (B) Single species and co-infections of embryos with uropathogenic *E. coli* (UPEC) strains NC_LB-5134 and LB-4801. * $P \leq 0.05$.

plasmids belonging to the FIB incompatibility group often contain virulence genes including aerobactin [31, 32] and are common in ExPEC strains such as UPEC [33] and APEC [34]. The genome annotations of the 4 selected ExPEC strains were manually searched to verify that all strains except LB-5658 contained the operon for conjugative plasmid transfer (Table 1). Interestingly, LB-5658 was the only strain in this study that grew less in mixed culture with and in proximity to EF and had no discernable phenotypic differences in the macrocolony proximity assay (Figs 1 and 5). Curing NC_LB-5134 of its conjugative F plasmid abolished both the growth and EPS production response to EF (Fig 7). This change may be due to loss of the episomal aerobactin siderophore or the ability to produce conjugative pilin via the *tra* operon, as cell-to-cell pilin connections have been shown to stimulate CA production resulting in robust *E. coli* biofilms [35]. In previous studies loss and acquisition of F plasmids resulted in attenuation while acquisition of F plasmids increased virulence of APEC [36, 37]. Surprisingly, the plasmid-cured NC_LB-5134 mutant was not attenuated in the embryo lethality assay alone or during co-infection with EF in the present study (Fig 7D). Embryo co-infection with LB-5658 that lacks a plasmid and EF resulted in greater lethality (Fig 6B), which suggests augmentation of ExPEC virulence by EF via a plasmid-independent mechanism. This could be unrelated to the *in vitro* phenotypic differences attributed to plasmid carriage observed in this study (Fig 7). Other mechanisms mediated by ExPEC chromosomal and episomal features and co-infecting EF strain factors cannot be ruled out. Thus, it appears that ExPEC response to EF is complex and may be linked to iron acquisition, F pilus formation, and CA production, which are interrelated. Further studies are needed to fully unravel the mechanism for EF induced ExPEC growth and virulence.

In conclusion, we found that interactions between ExPEC and EF were not limited to specific EF genotypes but that certain ExPEC phenotype responses to EF were associated with siderophores, conjugative plasmids, and CA production. To link these genotypes with observed ExPEC phenotypes, we investigated growth and EPS production of representative

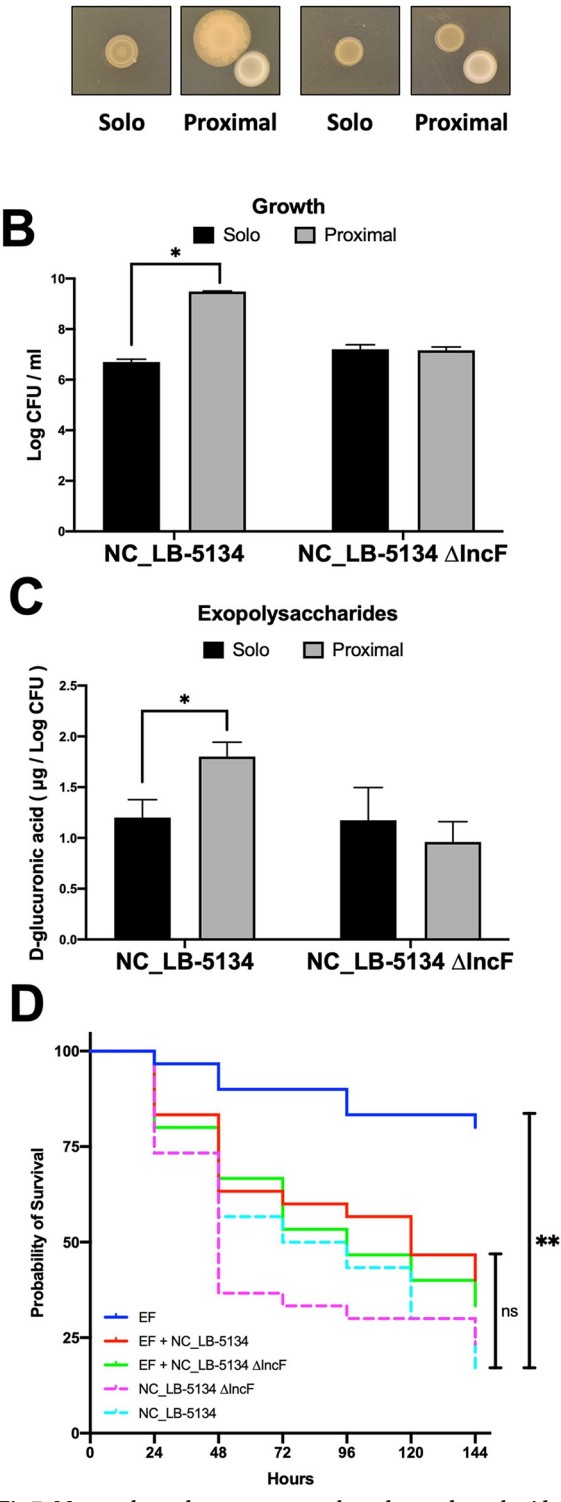

**Fig 7. Macrocolony phenotypes, growth, and exopolysaccharide (EPS) production by extraintestinal pathogenic *Escherichia coli* (ExPEC) strain NC_LB-5134 and plasmid-cured NC_LB-5134 ΔIncF.** (A) Macrocolony co-culture assay of NC_LB-5134 grown alone or 1 cm from *Enterococcus faecalis* OG1RF (EF, bottom right) after 5 d on trypticase soy agar with 10 mM glucose (TSA + G) and 0.15 mM iron chelator 2,2'-dipyridyl (22D). (B) Macrocolony growth of NC_LB-5134 and NC_LB-5134 ΔIncF from the co-culture assay. (C) Total EPS production of NC_LB-5134 and

plasmid-cured NC_LB-5134 ΔIncF, normalized to cell density. EPS was estimated by quantitating glucuronic acid using the method of Ma et al. (19). * $P \leq 0.05$. ** $P \leq 0.01$. (D) Survival of chicken embryos after challenge with ExPEC strains NC_LB-5134 and NC_LB-5134 ΔIncF alone or mixed with *Enterococcus faecalis* OG1RF (EF). In each experiment, 30 embryos were challenged with approximately $2 \times 10^2$ CFU/egg of ExPEC strains alone or mixed with $4 \times 10^3$ CFU of EF via the intra-allantoic route at 12 d of incubation and candled every 24 hr to monitor viability. * $P \leq 0.05$, ns = $P > 0.05$.

UPEC and APEC strains at both extremes of the phenotypic spectrum using a macrocolony proximity assay. EF augmented the growth and EPS production of ExPEC strains with IncFIB plasmid replicons and an intact CA biosynthesis operon while only growth was affected in a CA-deficient strain that harbored an F plasmid. These *in vitro* findings translated to enhanced virulence during co-infection of chicken embryos. Taken together, these results indicated that ExPEC respond to EF by increasing growth and production of EPS *in vitro*, and this phenotype is associated with IncF plasmid carriage and EPS biosynthesis genes. While additional studies are required to investigate the roles of plasmid and EPS production genes during the ExPEC response to EF and EF signals responsible for ExPEC EPS induction, the clinical importance of EF enhancement of ExPEC virulence in co-infection should be considered.

## Materials and methods

### Bacterial strains and growth conditions

APEC and EF clinical co-isolates were collected from diseased chickens and turkeys as described previously [5], sequenced, and deposited in GenBank under Bioprojects PRJNA293225 for APEC and PRJNA837978 for EF [38]. NC_LB-5134 and LB-4801 were previously designated as EC_06YS and EC_35YS, respectively [5]. Eleven UPEC were co-isolated with *Enterococcus* spp. from urine samples of dogs diagnosed with UTI, sequenced, and deposited in GenBank under Bioproject PRJNA293225 and are described elsewhere [39]. *Enterococcus faecalis* strain OG1RF [40] served as the EF strain for all experimental procedures. All strains were stored as glycerol stock cultures at -80˚C, revived on trypticase soy agar (TSA) with 5% sheep blood, and grown under static, aerobic conditions for 12 hr at 37˚C in Luria Bertani (LB) for *E. coli* or brain heart infusion (BHI) broth for EF prior to rinsing and normalization in PBS to OD$_{600nm}$ 0.4 or 0.7, respectively, which corresponded to approximately $2 \times 10^8$ CFU/ml for each strain.

### Macrocolony growth and proximity assays

Mixed-species growth and single-species macrocolony proximity assays followed methods of Keogh et al. (2016), with minor modifications [8]. For the mixed species growth assay, normalized suspensions of ExPEC and EF from above were used to prepare $1_{ExPEC}$:$19_{EF}$ mixed-species cultures or a $1_{ExPEC}$:$19_{PBS}$ single-species culture for the strain combinations. Three technical replicates for each of 11 UPEC strains in single-species or mixed-species culture were plated in 5 μl aliquots onto TSA containing 10 mM glucose (TSA + G) and 1.0 mM of the iron chelator 2,2'-dipyridyl (22D) (Sigma-Aldrich, St. Louis, MO, USA). After 24 h of incubation at 37˚C, UPEC growth was determined by excising the agar portion containing the macrocolony, resuspending cells from the section in 1 ml of PBS by vigorous mixing, and serially diluting and plating on MacConkey agar. For the macrocolony proximity assay, single-species ExPEC and EF cultures were plated on TSA + G with 0.15 mM 22D at 4 incremental, increasing distances from one another ranging from 1 to 7 cm on square 100 mm by 100 mm grid petri dishes for phenotype visualization. For growth and EPS quantitation experiments, ExPEC strains were plated either alone or 1 cm from EF on TSA + G with 0.15 mM 22D in replicant quadrants of

divided petri dishes and macrocolonies were excised and resuspended in sterile, distilled water instead of PBS as described above. Using the mixed-species growth assay, 4 UPEC and APEC co-isolated strains were classified as having high or low growth response phenotypes to EF and utilized for comparative genomic analysis: UPEC strains LB-5660 (robust growth in presence of EF) and LB-5658 (reduced growth in presence of EF) and APEC strains NC_LB-5134 (robust growth in presence of EF) and LB-4801 (reduced growth in presence of EF).

## Genetic characterization and comparative genomic analysis

Strain sequencing and *de novo* draft genome assembly methods used for strains in this study are described elsewhere [38, 39]. Clermont phylotyping of draft genome assemblies was conducted with the EZClermont web-based tool [41]. O-antigen and MLST typing were conducted with SerotypeFinder version 2.0 [42] and MLST version 2.0 [43] from the Center for Genomic Epidemiology, respectively. Plasmids were detected with PlasmidFinder version 2.1 with a preset identity threshold of 98% [44]. All detected plasmid replicons, regardless of identity, are provided as supplemental data (S2 Table). ExPEC and EF phylogenetic analyses were conducted with a single nucleotide polymorphism (SNP)-calling pipeline for WGS, CSIPhylogeny version 1.4 [45] with a minimum Z-score of 1.96, heterozygous SNPs ignored, and the altered FastTree options selected. The reference genomes UTI89 (NCBI accession NC_007946.1), APECO1 (NCBI accession: NC_008563.1), and *E. faecalis* 39EA1 (NCBI accession NZ_KZ846041.1) were used for SNP mapping for UPEC, APEC, and EF strains, respectively, and were included in the final dendrograms. Dendrograms were visualized using FigTree version 1.4.4 (http://tree.bio.ed.ac.uk/software/figtree/). The genome assemblies of LB-5660, LB-5658, NC_LB-5134 and LB-4801 were uploaded to the RAST (Rapid Annotation using Subsystems Technology) web package for functional annotation and comparative genomics [46]. Sequence-based comparisons were made between the two UPEC strains and the two APEC strains using the APECO1 reference genome annotation available in the RAST public database. Predicted proteins specific to ExPEC with augmented growth in response to EF were identified by filtering annotated genes with 98% similarity to reference sequences in the high response groups against genes that were undetectable in the low response groups. Unless otherwise specified, default parameters were used for all analyses.

## Exopolysaccharide quantitation

Total EPS was extracted from ExPEC macrocolonies to measure the glucuronic acid concentration of EPS using previously described methods [19]. Briefly, 800 μl of the macrococolony resuspension was mixed with 100 μl 1% Zwittergent (Sigma-Aldrich, St. Louis, MO, USA) in 100 mM citric acid (pH 2.0), incubated at 50˚C for 20 min, and centrifuged for 5 min at 16,000 *g*. A 250 μl aliquot of the supernatant was precipitated with 1 ml absolute ethanol for 30 min at 4˚C to separate EPS followed by an additional centrifugation for 5 min. After the resulting pellet was dried and dissolved in 200 μl deionized water, 1.2 ml of 12.5 mM borax in sulfuric acid (99.9%, Sigma-Aldrich, St. Louis, MO, USA) was added and mixed. The samples were then boiled for 5 min and allowed to cool to room temperature. Finally, 20 μl of 0.15% 3-hydroxydiphenol (Sigma-Aldrich, St. Louis, MO, USA) was added and mixed prior to measuring absorbance at 520 nm. Total glucuronic acid (μg/ml) for each sample was calculated based on a known standard curve and the relative EPS abundance was determined by normalizing this value with known Log CFU/ml values from the starting suspension to obtain μg of glucuronic acid/Log CFU.

## Embryo lethality assay

Animal experiments were approved by the Institutional Animal Care and Use Committee at North Carolina State University and performed in accordance with the guidelines and regulations for vertebrate embryos. Fertile broiler eggs were obtained from the NC State University Chicken Education Unit (Raleigh, NC). Parent broiler flocks were not vaccinated for *E. coli*. Eggs were incubated and infected using an embryo co-infection model for ExPEC and EF as described previously [5]. For each group, 30 to 35 eggs were candled at 11 d to verify viability and inoculated at 12 d with ExPEC and EF culture suspensions normalized in PBS as described above. Mixed-species infection groups received approximately $2 \times 10^2$ CFU of ExPEC and $4 \times 10^3$ CFU of EF_OG1RF, which was consistent with the *in vitro* mixed-species growth assay ratio of $1_{ExPEC}:19_{EF}$. Single species infection groups received $2 \times 10^2$ CFU of ExPEC or $4 \times 10^3$ CFU of EF_OG1RF. Eggs were candled every 24 hr for 5 d to determine viability. Allantoic fluid of all embryos, at time of death or termination of the study, was aerobically cultured to verify the presence of inoculated species in pure or mixed culture, which were differentiated by phenotype on TSA with 5% sheep blood. A negative control group of eggs injected with PBS only was included in each experiment. Allantoic fluid from the negative control eggs was cultured at the termination of each experiment and verified to be culture negative.

## Plasmid curing

*E. coli* strain NC_LB-5134 was cured of its F plasmid, pNC_LB-5134 encoding the *tra* genes and *tetA* for tetracycline resistance, using previously described methods [47]. Briefly, NC_LB-5134 was grown for three passages in LB broth with 10% sodium dodecyl sulfate at 37˚C with gentle shaking for 12 hr. Following serial dilution and plating on LB agar, colonies were replica patch plated onto LB agar and LB agar supplemented with 20 μg/ml tetracycline to identify tetracycline-sensitive isolates. Tetracycline-sensitive isolates were screened by PCR amplification of episomal virulence genes *hlyF* and *iss* [30]. Successful plasmid curing was verified by uploading the draft genome assembly of NC_LB-5134 ΔpNC_LB-5134 to PlasmidFinder version 2.1 [44] as described above with a return of no hits with the least stringent parameter settings (50% threshold for minimum identity and 20% minimum coverage).

## Statistical analysis

All CFU/ml values were log transformed. Differences between samples were determined using an unpaired t test and the two-stage step-up false-discovery rate for multiple comparisons [48]. Embryo survivability was plotted as Kaplan–Meier survival curves and differences determined by the log rank test of significance. All analyses were conducted with GraphPad Prism version 9.0 software. Unless otherwise stated, default parameters were used for all utilized software.

## Supporting information

**S1 Table. Predicted proteins present in responsive *E. coli* strains NC_LB-5134 and LB-5660.**
(XLSX)

**S2 Table. Plasmid replicons of the four *E. coli* strains investigated in this study.**
(XLSX)

## Author Contributions

**Conceptualization:** Grayson K. Walker, M. Mitsu Suyemoto, Megan E. Jacob, Siddhartha Thakur, Luke B. Borst.

**Data curation:** Grayson K. Walker.

**Formal analysis:** Grayson K. Walker.

**Funding acquisition:** Grayson K. Walker, Siddhartha Thakur, Luke B. Borst.

**Investigation:** Grayson K. Walker, Luke B. Borst.

**Methodology:** Grayson K. Walker.

**Project administration:** Luke B. Borst.

**Resources:** Megan E. Jacob, Siddhartha Thakur.

**Supervision:** M. Mitsu Suyemoto, Megan E. Jacob, Siddhartha Thakur, Luke B. Borst.

**Validation:** Grayson K. Walker, M. Mitsu Suyemoto, Luke B. Borst.

**Visualization:** Grayson K. Walker.

**Writing – original draft:** Grayson K. Walker, M. Mitsu Suyemoto, Luke B. Borst.

**Writing – review & editing:** Grayson K. Walker, M. Mitsu Suyemoto, Megan E. Jacob, Siddhartha Thakur, Luke B. Borst.

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
