## [Decision Letter · Decision Letter 0]

21 Aug 2024

PONE-D-24-07598Canine uropathogenic and avian pathogenic Escherichia coli harboring conjugative plasmids exhibit augmented growth and exopolysaccharide production in response to Enterococcus faecalisPLOS ONE

Dear Dr. Walker,

Thank you for submitting your manuscript to PLOS ONE. After careful consideration, we feel that it has merit but does not fully meet PLOS ONE’s publication criteria as it currently stands. Therefore, we invite you to submit a revised version of the manuscript that addresses the points raised during the review process.

We look forward to receiving your revised manuscript.

Kind regards,

Kwame Kumi Asare, Ph.D

Academic Editor

PLOS ONE

“The whole-genome sequencing was completed by the U.S. Food and Drug Administration (FDA) GenomeTrakr program-funded grant 1U18FD00678801. Additional funding was provided by the United States Department of Agriculture (USDA) Animal Plant Health Inspection Service National Bio and Agro-Defense Facility Scientist Training Program.”

“The whole-genome sequencing was completed by the U.S. Food and Drug Administration (FDA) GenomeTrakr program-funded grant 1U18FD00678801. Additional funding was provided by the United States Department of Agriculture (USDA) Animal Plant Health Inspection Service National Bio and Agro-Defense Facility Scientist Training Program. The views expressed in this article are the authors’ views and do not reflect official policies of the FDA, USDA, or the U.S. government. Reference to any commercial materials, equipment, or process does not in any way constitute approval, endorsement, or recommendation by the FDA or USDA.”

“The whole-genome sequencing was completed by the U.S. Food and Drug Administration (FDA) GenomeTrakr program-funded grant 1U18FD00678801. Additional funding was provided by the United States Department of Agriculture (USDA) Animal Plant Health Inspection Service National Bio and Agro-Defense Facility Scientist Training Program.”

Reviewers' comments:

Reviewer's Responses to Questions

**Comments to the Author**

1. Is the manuscript technically sound, and do the data support the conclusions?

Reviewer #1: Yes

Reviewer #2: Yes

2. Has the statistical analysis been performed appropriately and rigorously? 

Reviewer #1: Yes

Reviewer #2: Yes

3. Have the authors made all data underlying the findings in their manuscript fully available?

Reviewer #1: Yes

Reviewer #2: Yes

4. Is the manuscript presented in an intelligible fashion and written in standard English?

Reviewer #1: Yes

Reviewer #2: Yes

5. Review Comments to the Author

Reviewer #1: Check that abbreviations are used correctly throughout the manuscript. Should be written out the first time it is mentioned, and thereafter the abbreviation should be used.

The discussion is quite long and contains information already written in the results part. I would suggest that the authors try to limit the repetition of results and shorten the discussion to increase readability.

Reviewer #2: Answer: Within the framework One Health for All concept, this study of extraintestinal pathogenic E. coli (ExPEC), isolated from different hosts, is relevant and important. Overall, this is an interesting and well-written report, but some minor corrections are needed.

6. PLOS authors have the option to publish the peer review history of their article (what does this mean?). If published, this will include your full peer review and any attached files.

Reviewer #1: **Yes: **Hami Kaboosi

Reviewer #2: No

---

## [Author Response · Author response to Decision Letter 0]

14 Sep 2024

PONE-D-24-07598 Response to Reviewers

Reviewer #1: 

Check that abbreviations are used correctly throughout the manuscript. Should be written out the first time it is mentioned, and thereafter the abbreviation should be used.

AU. Thank you for pointing this out. We have ensured that abbreviations are only defined the first time they are presented in the text and only appear again in standalone figures.

The discussion is quite long and contains information already written in the results part. I would suggest that the authors try to limit the repetition of results and shorten the discussion to increase readability.

AU. We have minimized redundant information in the discussion section. We note, however, that brief reiteration of results is a necessary prelude to discussion. If there are specific areas where you feel results have been unnecessarily repeated, please specify these line numbers in the revised manuscript. 

Reviewer #2: 

Within the framework One Health for All concept, this study of extraintestinal pathogenic E. coli (ExPEC), isolated from different hosts, is relevant and important. Overall, this is an interesting and well-written report, but some minor corrections are needed.

AU. Thank you for acknowledging the relevance of our study and for your time spent reviewing the article. We note that you have indicated minor corrections are needed. In addition to making formatting changes in accordance with the PLOS ONE style guide, we have corrected redundant abbreviations and shortened the discussion. If there are specific areas where you feel additional corrections are needed, please specify these line numbers in the revised manuscript.

---

## [Decision Letter · Decision Letter 1]

14 Oct 2024

Canine uropathogenic and avian pathogenic Escherichia coli harboring conjugative plasmids exhibit augmented growth and exopolysaccharide production in response to Enterococcus faecalis

PONE-D-24-07598R1

Dear Dr. Walker,

We’re pleased to inform you that your manuscript has been judged scientifically suitable for publication and will be formally accepted for publication once it meets all outstanding technical requirements.

Kind regards,

Kwame Kumi Asare, Ph.D

Academic Editor

PLOS ONE

Additional Editor Comments (optional):

Reviewers' comments:

Reviewer's Responses to Questions

**Comments to the Author**

1. If the authors have adequately addressed your comments raised in a previous round of review and you feel that this manuscript is now acceptable for publication, you may indicate that here to bypass the “Comments to the Author” section, enter your conflict of interest statement in the “Confidential to Editor” section, and submit your "Accept" recommendation.

Reviewer #1: All comments have been addressed

Reviewer #2: (No Response)

2. Is the manuscript technically sound, and do the data support the conclusions?

Reviewer #1: Yes

Reviewer #2: Yes

3. Has the statistical analysis been performed appropriately and rigorously? 

Reviewer #1: Yes

Reviewer #2: Yes

4. Have the authors made all data underlying the findings in their manuscript fully available?

Reviewer #1: Yes

Reviewer #2: Yes

5. Is the manuscript presented in an intelligible fashion and written in standard English?

Reviewer #1: Yes

Reviewer #2: Yes

6. Review Comments to the Author

Reviewer #1: In the revised manuscript author(s) give satisfactory answer for raised queries. Corrections accepted. I appreciate the privilege

Reviewer #2: 1) In this study it was noted that “mortality did not increase with coinfection of EF and LB-5660 or NC_LB-5134 compared with a single-species infection with these two responsive E. coli strains“. In this regard, the question arose – how different were the profiles of virulence-associated genes (VAGs) for the 4 studied strains, which virulence genes were present in each of them according to the results of whole genome sequencing? It is recommended to supplement Table 1 with data on VAGs (or mention in the text that VAGs, for example, toxins, adhesines, have not been detected) because they are important for interpreting the results of in vivo experiments.

2) Mortality did not increase when coinfected with EF and LB-5660 or NC_LB-5134 compared to single-species infection with these two reacting E. coli strains. That is, there was no advantage coinfected with E. faecalis in vivo whereas in vitro an advantage in growth and exopolysaccharide production has been shown. Interestingly, the study (Karunarathna R. et al. Exposure of embryonating eggs to Enterococcus faecalis and Escherichia coli potentiates E. coli pathogenicity and increases mortality of neonatal chickens. Poult Sci. 2022;101(8):101983. doi:10.1016/j.psj.2022.101983. E) shows that “Although E. faecalis and E. coli alone did not affect the viability of embryos, a significantly high neonatal chicken mortality (27%) was observed following exposure of embryos to both E. faecalis and E. coli". There is no contradiction?

7. PLOS authors have the option to publish the peer review history of their article (what does this mean?). If published, this will include your full peer review and any attached files.

Reviewer #1: **Yes: **Hami Kaboosi

Reviewer #2: No

---

## [Editor Report · Acceptance letter]

28 Oct 2024

PONE-D-24-07598R1 

PLOS ONE

Dear Dr. Walker, 

I'm pleased to inform you that your manuscript has been deemed suitable for publication in PLOS ONE. Congratulations! Your manuscript is now being handed over to our production team.

Kind regards, 

on behalf of

Dr. Kwame Kumi Asare 

Academic Editor

PLOS ONE